# Coherent transfer of electron spin correlations assisted by dephasing noise

Takashi Nakajima [1], Matthieu R. Delbecq[1], Tomohiro Otsuka [1,2,3], Shinichi Amaha[1], Jun Yoneda [1], Akito Noiri [1,4], Kenta Takeda [1], Giles Allison[1], Arne Ludwig [5], Andreas D. Wieck [5], Xuedong Hu[1,6], Franco Nori [1,7,8] & Seigo Tarucha[1,4]

Quantum coherence of superposed states, especially of entangled states, is indispensable for many quantum technologies. However, it is vulnerable to environmental noises, posing a fundamental challenge in solid-state systems including spin qubits. Here we show a scheme of entanglement engineering where pure dephasing assists the generation of quantum entanglement at distant sites in a chain of electron spins confined in semiconductor quantum dots. One party of an entangled spin pair, prepared at a single site, is transferred to the next site and then adiabatically swapped with a third spin using a transition across a multi-level avoided crossing. This process is accelerated by the noise-induced dephasing through a variant of the quantum Zeno effect, without sacrificing the coherence of the entangled state. Our finding brings insight into the spin dynamics in open quantum systems coupled to noisy environments, opening an avenue to quantum state manipulation utilizing decoherence effects.

[1] RIKEN Center for Emergent Matter Science, Wako, Saitama 351-0198, Japan. [2] JST, PRESTO, 4-1-8 Honcho, Kawaguchi, Saitama 332-0012, Japan. [3] Research Institute of Electrical Communication, Tohoku University, 2-1-1 Katahira, Aoba-ku, Aoba-ku, Sendai 980-8577, Japan. [4] Department of Applied Physics, University of Tokyo, 7-3-1 Hongo, Bunkyo-ku, Tokyo 113-8656, Japan. [5] Lehrstuhl für Angewandte Festkörperphysik, Ruhr-Universität Bochum, D-44780 Bochum, Germany. [6] Department of Physics, University at Buffalo, SUNY, Buffalo, NY 14260-1500, USA. [7] Theoretical Quantum Physics Laboratory, RIKEN Cluster for Pioneering Research, Wako, Saitama 351-0198, Japan. [8] Physics Department, University of Michigan, Ann Arbor, MI 48109, USA. Correspondence and requests for materials should be addressed to T.N. (email: nakajima.physics@icloud.com) or to S.T. (email: tarucha@ap.t.u-tokyo.ac.jp)

One important goal of quantum technologies is to utilize entangled states in isolated, well-defined systems in a controllable manner. Decoherence, which scrambles the correlation between entangled parties through the interaction with the environment, is a major enemy of those quantum technologies, including quantum computation. In semiconductor quantum dot (QD) devices, potential building blocks of spin-based quantum computers, a great deal of effort have been made to mitigate the decoherence by engineering[1,2], controlling[3], and measuring[4,5] the environmental noise sources.

The decoherence effect is particularly significant in entangling gate operations for spin qubits because they are implemented by electrically tuning exchange coupling, which makes these qubits sensitive to charge noise[6]. To prepare an entangled state between non-nearest-neighbor spins, for instance, one could initialize a spin-singlet state in a double quantum dot and repeat the SWAP operations[7] between one of the two spins and the next-nearest-neighbor spins. For fast and high-fidelity control, the inter-dot exchange coupling $J$ has to be sufficiently larger than the Zeeman energy gradient $\Delta B$, while a large $\Delta B$ is also favorable for addressable single-spin control[8–10]. Repeating the SWAP operations with large $J$ is, however, practically difficult because it demands precise electrical control of sub-nanosecond pulse trains and the coherence time is decreased by the enhanced sensitivity to charge noise. In the opposite limit of $J \ll \Delta B$, the qubits are insensitive to charge noise and therefore an entangled state, once created, evolves coherently until it is dephased by inhomogeneous broadening of the Zeeman energy. Despite difficulties in engineering entanglement manipulation, quantum entanglement is believed to play essential roles even in systems coupled to noisy environments, including certain biological organisms[11]. Indeed, it was shown that decoherence can be used as a resource to stabilize entanglement in an artificial system[12].

Here we consider shuttling an entangled state to distant sites through an artificial spin chain (Fig. 1a). By using a simple, linear ramp of the detuning energy in an array of quantum dots with a magnetic field gradient (Fig. 1b), a typical setup for addressable single-spin control, the system undergoes adiabatic transitions across multi-level energy crossings as shown in Fig. 1c. At each energy crossing, one party of an entangled spin pair is swapped with the next spin. Repeating a similar process one after another yields entanglement between distant sites without direct interaction. We find that the efficiency of the adiabatic swap is significantly enhanced by pure dephasing—decoherence with no energy dissipation, suppressing non-adiabatic transitions across those energy crossings. Counter-intuitively, the coherence of the entangled state is preserved during this process. This scheme is generally applicable to an array of more than three quantum dots. We demonstrate this scheme in a gate-defined GaAs/AlGaAs triple quantum dot (TQD) with thoroughly characterized energy levels. Formation of an entangled state between sites $a$ and $b$ is demonstrated by observing the coherent evolution of a superposed state (singlet-triplet oscillation), $|\psi\rangle = \alpha|\downarrow_a\uparrow_b\rangle + e^{-i\Delta B_{ab}t/\hbar}\beta|\uparrow_a\downarrow_b\rangle$, where the two entangled sites $a$ and $b$ can be identified by the oscillation frequency determined by the Zeeman energy difference, $\Delta B_{ab}/h$. Numerical simulations show that the adiabatic spin swap is assisted by strong dephasing noise, which can be interpreted as a manifestation of the quantum Zeno effect.

## Results

**Generation and detection of correlated spin states.** The charge configuration of the TQD is controlled by the gate biases applied to plunger gates and probed by a nearby rf-QD charge sensor[13]. With the number of electrons in each QD, $N_i$, set to $(N_1N_2N_3) =$

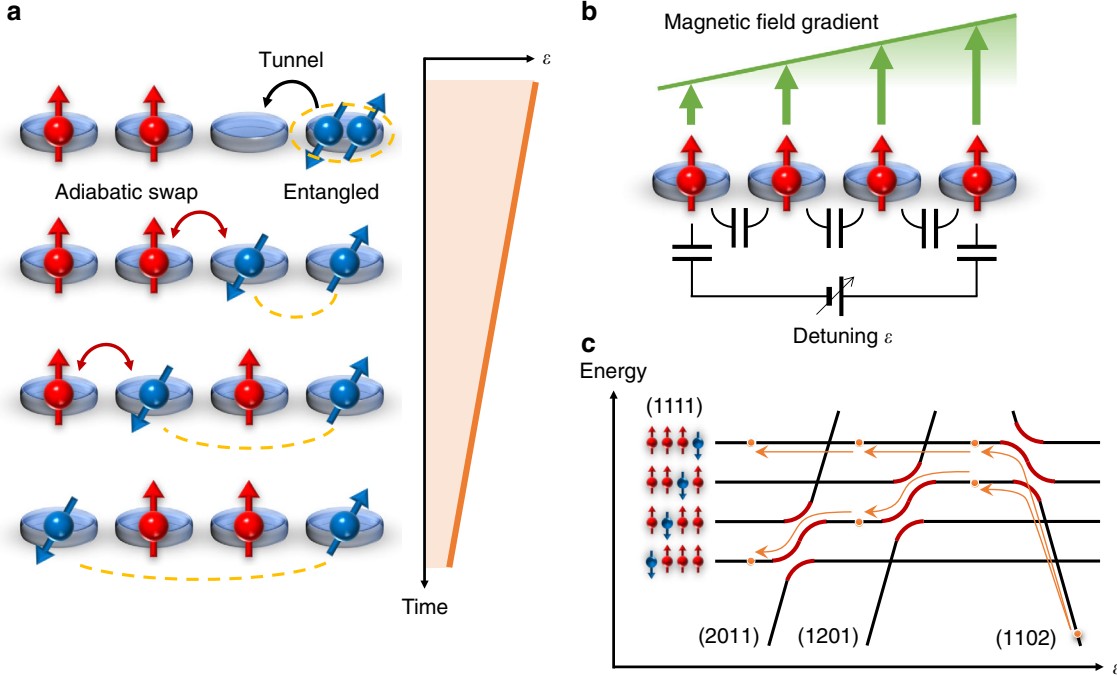

**Fig. 1** Entanglement shuttling in a spin chain. **a** Schematic of transferring an entangled state with a linear ramp of detuning energy $\varepsilon$ in an array of four spin qubits as an example. Note that the experiment described below is performed in an array of three spins, which is the minimum setup for demonstrating the concept. **b** Typical experimental setup of the spin array. The magnetic field gradient is prepared for addressable control of spin qubits by, e.g., a micromagnet[8–10]. A gate bias voltage applied between both ends of the array makes an electrostatic potential gradient via capacitive coupling. **c** Energy diagram corresponding to the setup shown in **b**. Such a configuration is realized when, e.g., the electrostatic energy differences between neighboring dots are equally modulated by $\varepsilon/3$ and each potential has a proper energy offset. Orange arrows show the adiabatic state transitions for the entanglement transfer and orange circles indicate two superposed states at each step

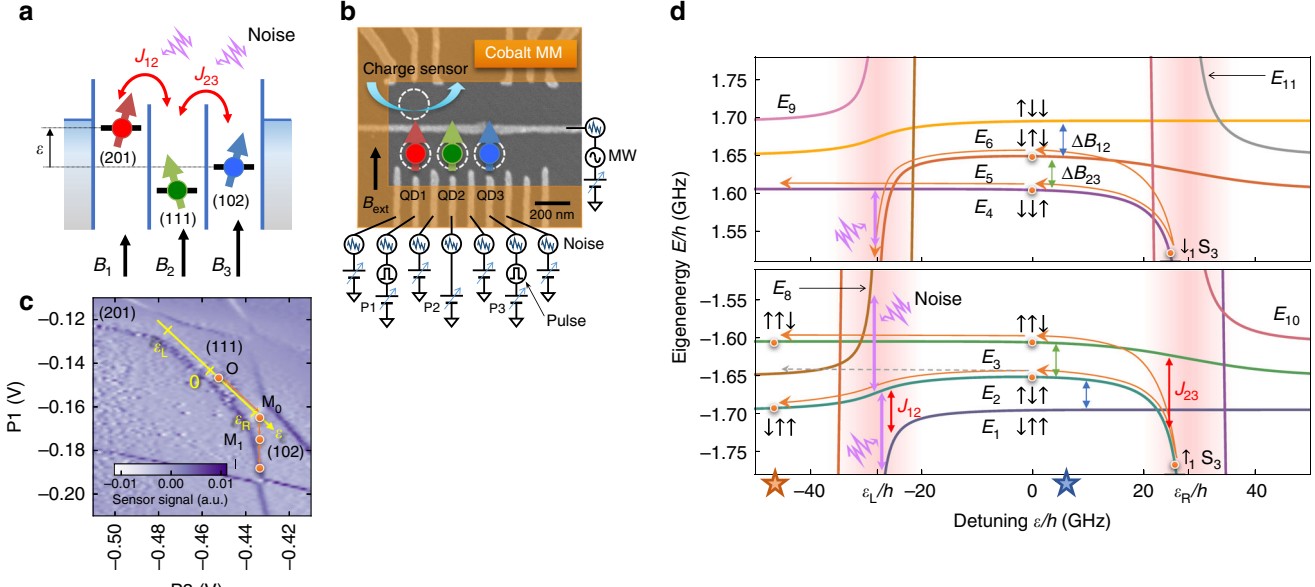

**Fig. 2** Experimental implementation of the spin chain. **a** Schematic of the spin chain with three sites studied in this work. **b** Schematic of a TQD device similar to the one measured, containing single-electron spins in each QD. The TQD is fabricated in a GaAs/AlGaAs heterostructure. The cobalt micromagnet deposited on a calixarene insulation layer is magnetized by an externally applied in-plane magnetic field of $B_{ext} = 0.7$ T and it induces a difference of the local Zeeman energy $\Delta B_{12}$ ($\Delta B_{23}$) between QD2 and QD1 (QD3 and QD2). The spin states are manipulated by DC gate biases, pulse voltages, microwave signals, and thermal noise applied on finger gate structures. **c** Charge stability diagram of the TQD obtained by differentiating the rf-reflectometry signal of the nearby QD charge sensor. The yellow arrow shows the detuning axis along which the potential energy detuning $\varepsilon$ between QD1 and QD3 is controlled by gate voltages P1 and P3. The bias points for spin initialization (I), operation (O), and measurements ($M_{0,1}$) are marked by circles. **d** Energy diagram of the three-spin states. The Hamiltonian parameters are extracted from the energy spectroscopy in Fig. 4. The energy levels for $S_z = -1/2$ and $+1/2$ are shown in the upper and lower panels, respectively (see Supplementary Fig. 3 for the $S_z = \pm 3/2$ branches). Stars indicate the detuning values at which we measure the coherent evolutions plotted in Fig. 3

(111), voltage pulses are superposed to the gate biases to control the energy offset between charge configurations (102) and (201) along the line shown in Fig. 2c. This allows us to control the detuning energy $\varepsilon$ between the left and right QDs while leaving the energy of the (111) configuration unchanged. A top cobalt micromagnet induces an inhomogeneous Zeeman field[8–10] along the externally applied in-plane magnetic field $B_{ext}$. By design, the transverse field component induced by the micromagnet is much smaller. The Zeeman energy difference $\Delta B_{ij}$ between QD$i$ and QD$j$ ($i,j = 1, 2, 3$) splits the degenerate three-spin states with the same z component of total spin to discrete levels $E_k$ (labeled from lower to higher energies at $\varepsilon = 0$ as shown in Fig. 2d). When the exchange energy $J_{ij}$ is negligible ($|\Delta B_{ij}/J_{ij}| \gg 1$), the three-spin eigenstates are described by $|\sigma_1 \sigma_2 \sigma_3\rangle = |\sigma_1\rangle \otimes |\sigma_2\rangle \otimes |\sigma_3\rangle$ ($\sigma_i = \uparrow, \downarrow$) rather than the quadruplet and doublets[14–16], allowing us to access individual spin states[17].

An entangled spin pair is locally prepared by initializing the system in (102), where two electrons in QD3 occupy the singlet ground state, $S_3$. The electron in QD1 is left uninitialized and its spin state is thermally populated with the Boltzmann distribution. Thus the system is initially in a mixed state described by the density matrix, $\rho_0 = r|\uparrow_1 S_3\rangle\langle\uparrow_1 S_3| + (1-r)|\downarrow_1 S_3\rangle\langle\downarrow_1 S_3|$, with $r \approx 0.7$ resulting from the electron temperature, $T_e = 240$ mK. The spin pair is then split to QD2 and QD3 by rapidly displacing $\varepsilon$ to zero in (111) (Methods section). When the displacement is slow enough compared to the inter-dot tunnel rate $t_R$, but rapid enough against $\Delta B_{23}$, the spin state is unchanged and $|S_3\rangle$ is loaded into a spin singlet between QD2 and QD3, $|S_{23}\rangle = \frac{1}{\sqrt{2}}(|\uparrow_2\downarrow_3\rangle - |\downarrow_2\uparrow_3\rangle)$ (as is the case for double QDs when QD1 is neglected). The loaded state starts to oscillate with time $t_{evolve}$

between $|S_{23}\rangle$ and $|T_{23}\rangle = \frac{1}{\sqrt{2}}(|\uparrow_2\downarrow_3\rangle + |\downarrow_2\uparrow_3\rangle)$ (spin triplet) as shown in Fig. 3b, reflecting the relative phase evolution between $|\downarrow_2\uparrow_3\rangle$ and $|\uparrow_2\downarrow_3\rangle$ due to the energy difference $\Delta B_{23}/h = 45$ MHz. The singlet fraction in the final state is read out by unloading $S_{23}$ back to $S_3$ in the reverse process. Hereafter, the probability of finding the singlet final state in (102) is denoted "singlet-return probability", $P_S$.

**Transfer of entangled states to a distant site.** The entanglement created in QD2 and QD3 is then transferred to the distant spin pair in QD1 and QD3 by pulsing $\varepsilon$ more negatively. A distinct feature of the coherent oscillation between $S_{13}$ and $T_{13}$ is visible in the region $\varepsilon \ll \varepsilon_L$, where the oscillation frequency is governed by the energy gap $\Delta B_{13} = \Delta B_{12} + \Delta B_{23}$ (red data points in Fig. 3b). This frequency is clearly distinguished from that of the $S_{23}$–$T_{23}$ precession governed by $\Delta B_{23}$. The generation of this distant entanglement could be understood by the adiabatic transition through a multi-level avoided crossing depicted in Fig. 2d. When $\varepsilon$ is swept from $\varepsilon = 0$ to $\varepsilon \ll \varepsilon_L$, $|\uparrow_1 \downarrow_2 \uparrow_3\rangle$ is adiabatically loaded into $|\downarrow_1\uparrow_2\uparrow_3\rangle$, following the eigenenergy marked as $E_2$. This process adiabatically swaps the spins in QD1 and QD2, transferring $|\uparrow_1 S_{23}\rangle$ to $|\uparrow_2\rangle \otimes |S_{13}\rangle = \frac{1}{\sqrt{2}}(|\uparrow_1\uparrow_2\downarrow_3\rangle - |\downarrow_1\uparrow_2\uparrow_3\rangle)$. On the other hand, the fraction of $\sigma_1 = \downarrow$ in $\rho_0$ is transferred to $|\downarrow_1 S_{23}\rangle$ at $\varepsilon = 0$ and then to $\frac{1}{\sqrt{2}}(|S_1 \downarrow_3\rangle - |\downarrow_1\downarrow_2\uparrow_3\rangle)$ at $\varepsilon < \varepsilon_L$ [as $|\downarrow_1\downarrow_2\uparrow_3\rangle$ is unloaded to $|S_1 \downarrow_3\rangle$, see the upper panel of Fig. 2d]. Since this state is a superposition of different charge sectors in (111) and (201), it decoheres immediately[6] and makes no contribution to the observed oscillations. However, the quantitative analysis presented below reveals that this naive interpretation is insufficient; as shown by the gray line in Fig. 3b obtained from a numerical simulation, only the $S_{23}$–$T_{23}$ precession with frequency $\Delta B_{23}/h$

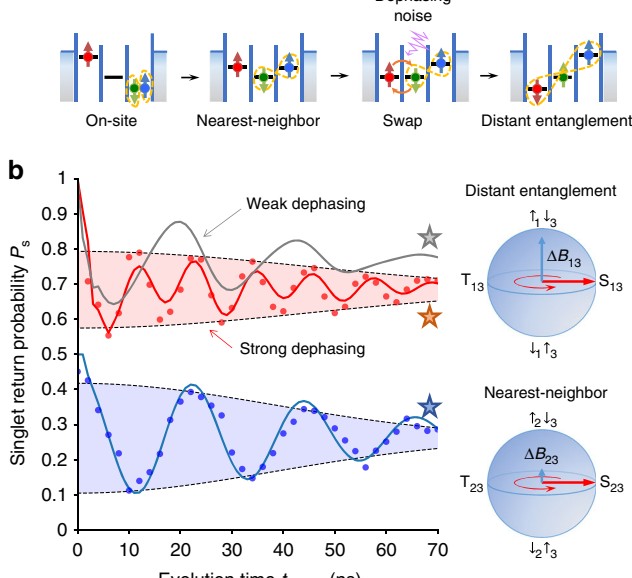

**Fig. 3** Coherent transfer of spin entanglement. **a** Illustration of the entanglement transfer process. Local spin entanglement is prepared in QD3 and then split to the nearest neighbors (QD2, QD3), followed by the noise-assisted transfer to distant sites (QD1, QD3). **b** Coherent evolutions of the distant entanglement (taken at $\varepsilon/h = -44$ GHz, red circles) and the nearest-neighbor entanglement ($\varepsilon/h = -7$ GHz, blue circles offset for clarity). The simulation data are scaled to take into account the readout error in the data (Methods section). The singlet-return probability $P_S$ inferred from the single-shot spin blockade measurement is plotted as a function of $t_{evolve}$. The data points are obtained by performing a Gaussian convolution filter of the width $\sigma_\varepsilon = 0.9$ GHz for the detuning. Solid lines show the numerical calculations of the coherent evolutions at $\varepsilon/h = -44$ GHz (red and gray) and $\varepsilon/h = -7$ GHz (blue) with the dephasing rates $\gamma_{t_L} = 1.7$ GHz, $\gamma_{t_R} = 0.12$ GHz and $\gamma_{\varepsilon_{L,R}}/\gamma_{t_{L,R}} = 100$ (red and blue) and with the smaller rates of $\gamma_{t_L} = 17$ MHz and $\gamma_{t_R} = 1.2$ MHz (gray). The simulation results are reproduced from Fig. 4c by choosing corresponding detuning values marked by stars. The envelopes of the oscillations correspond to a Gaussian decay with $T_2^* \approx 60$ ns for both cases due to the Overhauser field fluctuation[5] during the ensemble averaging time of 22.7 s. This phase averaging effect is independent of the Markovian dephasing noise which is dominant only around $\varepsilon = \varepsilon_{L,R}$ as discussed later

as the energy gap increases (decreases) for $\sigma_1 = \uparrow$ ($\downarrow$) due to $J_{12}$. Fitting the two spectral lines with the model calculations (red solid and blue dashed lines in Fig. 4b) allows us to extract the three-spin energy diagram without arbitrary parameters.

Although the observed oscillations are assigned to the coherent evolution between the energy levels discussed above, finite state leakage to other levels also happens. The relevant dynamics has been studied in double quantum dots[19,20]. First, the singlet $S_3$ prepared in QD3 may fail to tunnel to QD2 during the detuning sweep and the system stays in the (102) charge configuration ($E_{10}$ or $E_{11}$ energy branches). This state does not contribute to the coherent oscillations because the coherence with the states in the (111) charge sector is lost in a much shorter timescale. It contributes, however, to the increase of the mean value of $P_S$. Second, $S_3$ may be loaded to a spin-polarized triplet state $T_{+23} = |\uparrow_2\uparrow_3\rangle$ when the detuning crosses the energy resonance point. The position of the $S_3$–$T_{+23}$ resonance is identified by measuring the $S_3$–$T_{+23}$ mixing caused by the transverse magnetic field gradient. We find that this mixing rate is well below 10 MHz, as expected for the small transverse component (<10 mT) and the large external magnetic field ($B_{ext} = 0.7$ T). The state leakage to $T_{+23}$ is therefore negligible when the detuning is swept past the resonance with the ramp time of $t_s = 1.6$ ns used here (Methods section). For this reason, the transverse field component is neglected in the simulation.

**Analysis of the dephasing effect.** The coupling of the spin system to pure dephasing noise in the environment results in fluctuations of the eigenenergies $E_k$. Although the magnetic fluctuation of $\Delta B_{ij}$ due to randomly oriented nuclear spins is significant in the GaAs material, its influence is pronounced at a rather long-timescale, leading to an ensemble phase averaging effect[5,21]. Meanwhile, $E_k$ is also susceptible to the exchange noise in $J_{ij}$, which is in turn susceptible to the charge noise in ($\varepsilon - \varepsilon_{L,R}$) and $t_{L,R}$. This effect is especially significant when the derivatives of $E_k$ with those parameters are large[6] (red shaded region in Fig. 2d). Assuming that the noise sources are uncorrelated with each other, the fluctuation of $E_k$ can be represented by those of $\varepsilon_{L,R}$ and $t_{L,R}$ around $\varepsilon \approx \varepsilon_{L,R}$ without loss of generality. In the Markov approximation, this effect is described by the Lindblad operators

$$\mathcal{L}_{\varepsilon_{L,R}} = \sqrt{2\gamma_{\varepsilon_{L,R}}} \sum_k \frac{\partial E_k}{\partial \varepsilon_{L,R}} |k\rangle\langle k|, \quad \mathcal{L}_{t_{L,R}} = \sqrt{2\gamma_{t_{L,R}}} \sum_k \frac{\partial E_k}{\partial t_{L,R}} |k\rangle\langle k|,$$

(1)

where $|k\rangle$ is the energy eigenstate of $E_k$. The operators in this form lead to pure dephasing in a superposed state of $|k\rangle$ and $|l\rangle$ at the rate $\Gamma_{kl} = \gamma_{\varepsilon_{L,R}} \left(\frac{\partial E_{kl}}{\partial \varepsilon_{L,R}}\right)^2 + \gamma_{t_{L,R}} \left(\frac{\partial E_{kl}}{\partial t_{L,R}}\right)^2$, with $E_{kl} = E_k - E_l$. Based on these dephasing terms and the extracted energy diagram, we numerically simulate the coherent evolution of the system (Methods section). Figure 4c shows that the coherent oscillation of the distant entanglement, visible in Fig. 3b and in the region of $\varepsilon < \varepsilon_L$ in Fig. 4a, is entirely reproduced only when the dephasing rates estimated from the experiment are taken into account. We notice, however, that the adiabatic transition from $|\uparrow_1\downarrow_2\uparrow_3\rangle$ ($\varepsilon = 0$) to $|\downarrow_1\uparrow_2\uparrow_3\rangle$ ($\varepsilon < \varepsilon_L$) is not perfect in the simulation with our choice of the dephasing rates. This leads to a residual oscillatory component between $|\uparrow_1\downarrow_2\uparrow_3\rangle$ and $|\uparrow_1\uparrow_2\downarrow_3\rangle$ with frequency $\Delta B_{23}/h$ superposed onto the red line in Fig. 3b, which is not discernible in the experimental data. This discrepancy may be resolved by taking into account more accurate dephasing rates and noise spectrum. We thus demonstrate that the pure dephasing noise plays an essential role in performing the

would be observed if strong dephasing noise were absent. This is because the coherent Landau–Zener transition non-adiabatically brings $|\uparrow_1\downarrow_2\uparrow_3\rangle$ to the identical spin state in the energy branch marked as $E_8$, as the relevant energy splitting, $(2t_L/h)^2 \approx 0.4$ GHz$^2$, is much smaller than the Landau–Zener velocity determined by the pulsing of $\varepsilon$, $v_{LZ}/h \approx 60$ GHz$^2$.

**Three-spin state spectroscopy.** To simulate the coherent dynamics of the system, we extract the system energy diagram shown in Fig. 2d by measuring[2,18] coherent oscillations of $P_S$ at various values of $\varepsilon$. Figure 4a shows that the oscillation frequency changes with $\varepsilon$, representing the energy gaps between two pairs of superposed spin states ($|E_3 - E_2|$ and $|E_5 - E_4|$ marked by green arrows in Fig. 2d). The energy spectrum is mapped out by the 1D Fourier transform of this data as shown in Fig. 4b. A single-spectral peak at $\varepsilon \gtrsim 0$ represents that the energy gaps are almost unchanged around $\varepsilon \approx 0$, being solely given by $\Delta B_{23}/h = 45$ MHz since $J_{12}$ and $J_{23}$ are quenched, and increase equally with $J_{23}$ for more positive $\varepsilon$. On the other hand, the spectral lines split at $\varepsilon < 0$

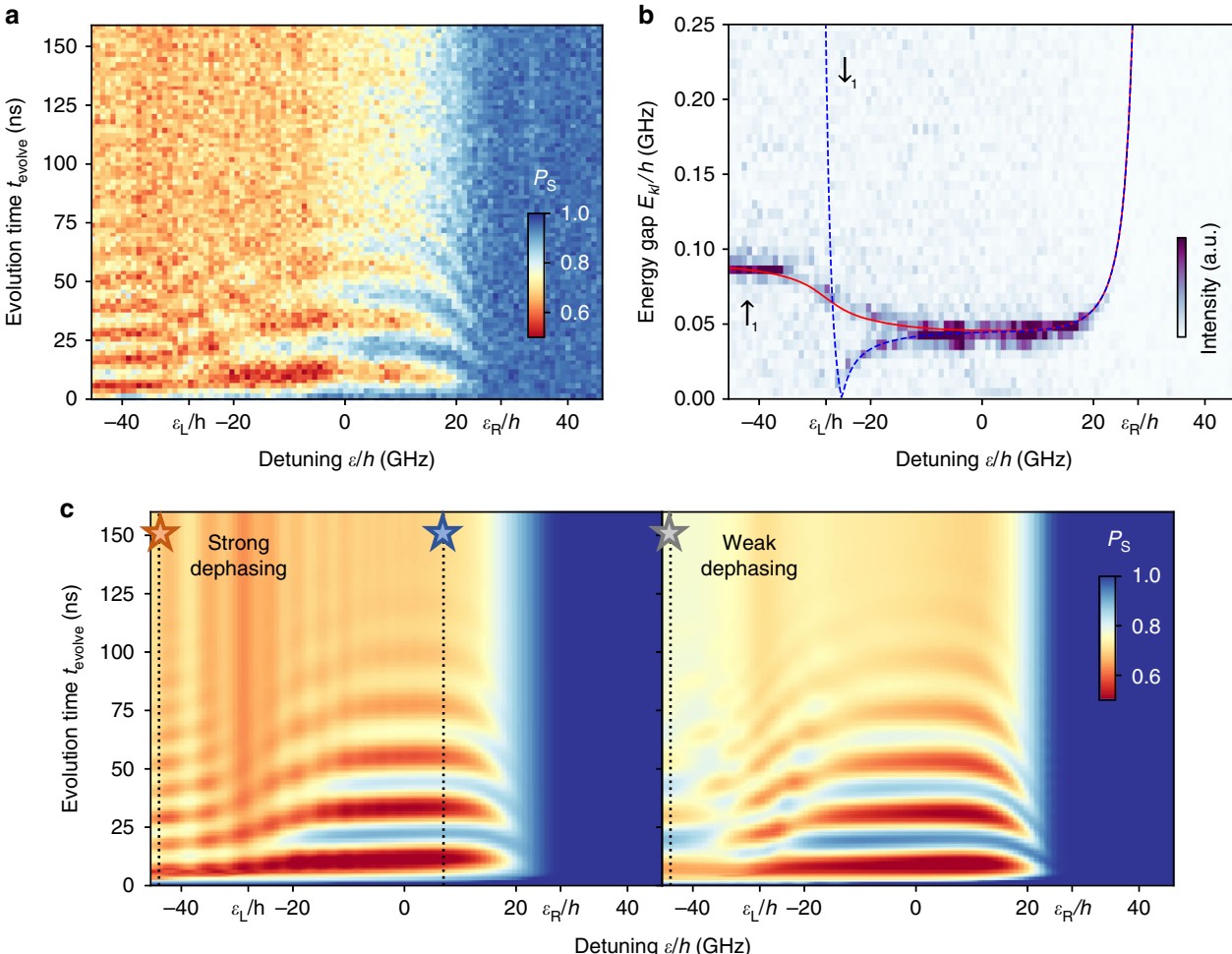

**Fig. 4** Coherent evolution and energy spectroscopy of the three-spin system. **a** Coherent evolution of the three-spin state taken at various values of $\varepsilon$ with the same measurement sequence as the one used in Fig. 3. **b** Fast Fourier transform of the data in **a** for each detuning value. A red solid line (blue dashed line) represents the fit to the energy gap between the two branches marked by $E_2$ and $E_3$ ($E_4$ and $E_5$) in Fig. 2b, which results from the fraction of $|\uparrow_1 S_3\rangle$ ($|\downarrow_1 S_3\rangle$) in $\rho_O$. Fitting allows us to extract the QD parameters as the inter-dot tunnel couplings $t_L/h = 0.30$ GHz and $t_R/h = 0.43$ GHz, the detuning values of (201)–(111) and (111)–(102) resonances $\varepsilon_L/h = -28$ GHz and $\varepsilon_R/h = 28$ GHz, and the local Zeeman field differences $\Delta B_{12}/h = 45$ MHz and $\Delta B_{23}/h = 45$ MHz in agreement with the electron spin resonance spectra (see Methods section and Supplementary Fig. 2a). **c** The numerical calculation of the coherent evolution in **a** performed with the dephasing rates of $\gamma_{t_L} = 1.7$ GHz and $\gamma_{t_R} = 0.12$ GHz (left) and with the rates of $\gamma_{t_L} = 17$ MHz and $\gamma_{t_R} = 1.2$ MHz (right)

adiabatic spin swap in QD1 and QD2 required for generating distant entanglement.

Before discussing the interpretation of the result, we check the validity of our noise model by a charge dephasing measurement[6] (Methods section). Figure 5a shows a typical damping curve of the exchange oscillation near $\varepsilon = \varepsilon_R$, being well characterized by an exponential decay rather than a Gaussian. This is consistent with the quasi-white spectrum and the Markovian nature of the noise[22], which is here attributed to the Johnson–Nyquist noise on gate bias voltages (Methods section). The decay time extracted from the envelope is plotted versus $\varepsilon$ in Fig. 5b. It is seen that the decay rate increases as $\varepsilon$ approaches $\varepsilon_{L,R}$ where the exchange noise dominates the magnetic fluctuation. The dephasing rate in this regime fits very well with $\Gamma_{kl}$ derived from Eq. 1, giving the values of $\gamma_{\varepsilon_{L,R}}$ and $\gamma_{t_{L,R}}$. This result clearly suggests that the dephasing rate overwhelms the relevant energy splittings, $\Gamma_{kl} \gg t_{L,R}/h$, such that the dephasing effect plays an essential role in the transition process[23].

## Discussion

The effect of the dephasing noise can be interpreted as the manifestation of the quantum Zeno effect, also known as the 'watchdog' effect, which forces a system to be in an eigenstate by frequent projective measurements[24]. More generally, the environment-induced dephasing can be viewed as a continuous monitoring of the system by the environment[23] as recently demonstrated experimentally[25]. This effect tends to force Landau–Zener transitions[26] to be adiabatic even when the Landau–Zener velocity is in the non-adiabatic regime. In this work, the transition between QD1 and QD2 would be non-adiabatic $[(2t_L)^2 \ll h\nu_{LZ}]$ in the absence of the dephasing noise, so that spin entanglement fails to propagate to QD1 as shown in the right panel in Fig. 4c. However, the strong dephasing noise $[\Gamma_{12}, \Gamma_{28} \gg t_L/h]$ keeps this process adiabatic, facilitating the adiabatic spin swap between QD1 and QD2. This effect is significantly enhanced around $\varepsilon \approx \varepsilon_L$, where superposed states in $E_1$ and $E_2$ or in $E_2$ and $E_8$ are sensitive to the charge noise due to the large derivatives $|\partial E_{kl}/\partial \varepsilon_{L,R}|$ and $|\partial E_{kl}/\partial t_{L,R}|$. The resulting dephasing with rates $\Gamma_{12}$ and $\Gamma_{28}$ keeps the eigenstate of $E_2$ staying in the instantaneous energy eigenstate throughout the passage across $\varepsilon = \varepsilon_L$. It is noteworthy that the coherent evolution of the generated entangled state is observed even in the presence of the strong dephasing noise utilized to generate it. This is because the entangled state is in the decoherence-free subspace[27–29] spanned

**a**

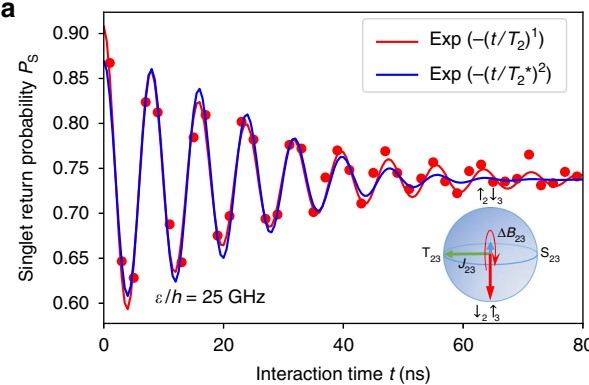

**b**

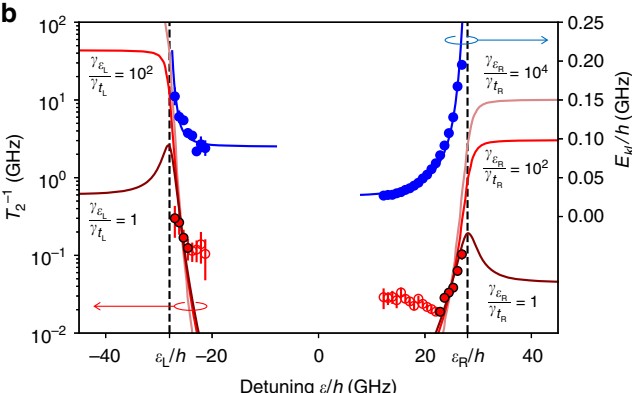

**Fig. 5** Markovian dephasing measurement of the three-spin system. **a** Exchange oscillation (SWAP) between $|\sigma_1 \uparrow_2\downarrow_3\rangle$ and $|\sigma_1 \downarrow_2\uparrow_3\rangle$ driven by a brief excursion to $\varepsilon/h = 25$ GHz for the duration of $t$, following the adiabatic loading of $|\sigma_1 \downarrow_2\uparrow_3\rangle$ (Methods section). Red and blue lines are the fits to the decaying oscillations $A\exp\left[-\left(t/T_2^{(*)}\right)^a\right]\cos(Et/\hbar + \phi) + B$, with exponential ($a = 1$ with the dephasing time $T_2$) and Gaussian ($a = 2$ with the emsemble dephasing time $T_2^*$) envelopes, respectively. **b** The dephasing rate $T_2^{-1}$ (red circles) and the oscillation frequency $E_{kl}/h$ (blue circles) extracted from the fits as in **a** with the exponential envelope at various $\varepsilon$ with the error bars representing the standard errors of the fitting. Blue lines represent energy gaps for $E_{21}$ and $E_{32}$ calculated with the same parameters as those used in Figs. 2 and 4 except for $\Delta B_{12}/h = 90.5$ MHz and $\Delta B_{23}/h = 26.7$ MHz determined by the fitting. Those values differ from the ones derived in Fig. 4 due to the slow Overhauser field fluctuation. Red lines are the dephasing rates $\Gamma_{12}$ and $\Gamma_{23}$ arising from the charge noise modeled by Eq. 1 for different magnitudes of $\gamma_{\varepsilon_{L,R}}/\gamma_{t_{L,R}}$. The values of $\gamma_{t_L}$ and $\gamma_{t_R}$ used in Figs. 3b and 4c are derived from the fitting by choosing $\gamma_{\varepsilon_{L,R}}/\gamma_{t_{L,R}} = 100$. The data points with red empty circles deviate from the model because the fluctuation of $\Delta B_{ij}$ also contributes to the Gaussian decay and a fitting with an exponential envelope is unreliable in the corresponding detuning range

by $E_3$ and $E_2$. This subspace is effectively decoupled from the identical noise sources thanks to much smaller $|\partial E_{32}/\partial \varepsilon_{L,R}|$ and $|\partial E_{32}/\partial t_{L,R}|$. We have thus demonstrated that, if properly used, the dephasing effect can enhance the manifestation of purely quantum mechanical properties in a noisy environment.

Finally, we comment on the applicability of the present technique to a larger array of quantum dot spins. The generation of a distantly entangled state by a single-adiabatic detuning pulse (Fig. 1) is technically much easier to implement than repeating the conventional SWAP operations by switching the exchange coupling non-adiabatically for calibrated durations. On the other hand, the detuning pulse must be sufficiently slow to realize the adiabatic limit, leading to slower control in a larger array. This requirement is partly alleviated by the use of dephasing noise as

shown in the present work. However, the fidelity of the process is limited by the lack of controllability of the dephasing noise. We expect that control of dephasing noise by, e.g., external noise generators should allow faster and more reliable entanglement shuttling.

In conclusion, the noise-assisted transfer of spin correlations demonstrated in our experiment may open the possibility for arbitrary relocation of quantum entanglement and using its quantum nature in larger scale devices. The underlying physics may also be relevant to the possible manifestation of quantum entanglement in biological systems, such as photosynthetic light-harvesting complexes and avian chemical compasses[11]. Our results suggest that the QD devices can be used as a powerful platform for quantum simulation of the open quantum systems coupled to noisy environments.

## Methods

**Measurement.** The pulse sequences used in the experiment, illustrated in the Supplementary Fig. 1, were generated by a Tektronix AWG70002 arbitrary waveform generator operated at 1 GSa per s. The ouput waveform was low-pass filtered by Mini-Circuits SBLP-300+ to adjust the rise time of the rapid adiabatic pulse used in the experiment, as well as to filter out high-frequency noise. We find that the step response of the filter is well approximated by using the error function as $\psi(t) = \frac{1}{2}\left[1 + \mathrm{erf}\left(\frac{t - t_0}{t_s}\right)\right]$ with $t_s = 1.6$ ns. This effect is taken into account in the numerical calculations described below. The pulse signals are fed to the sample through broadband coaxial cables in the dilution refrigerator with their $-3$ dB point at around 10 GHz. The signals are attenuated by 9 dB in total through cryogenic attenuators installed at each cold plate.

At the beginning of each pulse cycle, the spins in QD2 and QD3 are initialized to the doubly occupied singlet state $S_3$. For the entanglement measurement in Figs. 3b and 4a, this is achieved at point $M_0$, where the relaxation of the triplet state with anti-parallel spins, $T_{23}$, is fast because of the rapid state mixing with $S_{23}$ by $\Delta B_{23}$ and efficient phonon emission[30]. The state $|\sigma_1 S_3\rangle$ is then rapidly loaded[31,32] into (111) by setting the detuning to the target value $\varepsilon$ at point O within the rise time of $t_s$. In this rapid adiabatic passage process, $|\sigma_1 S_3\rangle$ remains in the singlet state $|\sigma_1 S_{23}\rangle$ as far as $J_{12}$ is negligible compared to $\Delta B_{12}$. After the state evolution for $t_{\mathrm{evolve}}$, the detuning is pulsed back for readout and $|\sigma_1 S_{23}\rangle$ returns to $|\sigma_1 S_3\rangle$ while all the other triplet components remain spin-blocked. The readout is performed at point $M_1$ close to the triplet resonance point, where the relaxation of $T_{23}$ is suppressed by the exchange coupling. The final charge configuration is probed by integrating the sensor signal for 4 μs and the outcome is mapped to either the singlet or one of the triplets[13,33]. To cancel out the slow drift of the background signal, the charge sensor signal in (102) is recorded before every spin manipulation at point O and it is subtracted from the integrated signal at $M_1$. The whole-pulse cycle is repeated for $N = 200$ times to infer the singlet-return probability[13,33] $P_S$.

During the acquisition of the data in Figs. 3 and 4a, the spectrum of $\Delta B_{ij}$ is broadened by the Overhauser field fluctuation[5,21]. To make the fluctuation spectrum homogeneous for the whole-data set in a single acquisition, independent variables ($t_{\mathrm{evolve}}$ and $\varepsilon$) are scanned consecutively to obtain single samples for each point and the whole scan is repeated $N = 200$ times. This is essential to distinguish oscillation frequencies obtained at different values of $\varepsilon$ without being disturbed by the fluctuation of $\Delta B_{ij}$. The singlet-return probability $P_S$ is inferred from the ensemble average for each set of independent variables. The whole-pulse cycle is finished in 22.7 s, which is shorter than the typical nuclear spin decorrelation time. This allows us to obtain a moderate inhomogeneous dephasing time of $T_2^* \approx 60$ ns, though it also brings about a drift in $\Delta B_{ij}$ from measurement to measurement[5].

The dephasing measurement in Fig. 5 is performed in a similar pulse cycle but with the two-spin 'SWAP' pulse sequence shown in Supplementary Fig. 1b. After the system is initialized at $M_0$, the detuning is displaced rapidly to cross the $S_3$–$T_{+23}$ resonance line, and then ramped slowly (within 1 μs) to $\varepsilon = 0$. During this slow passage, $|\sigma_1 S_3\rangle$ is loaded into the eigenstate of the local Zeeman field, $|\sigma_1 \downarrow_2\uparrow_3\rangle$. The detuning is then positively (negatively) displaced to turn the exchange coupling $J_{23}$ ($J_{12}$) on. During the hold time of $\tau_{\mathrm{SWAP}}$, spins in QD2 and QD3 (QD1 and QD2) are swapped at frequency $J_{ij}(\varepsilon)/h$. Readout is performed after the detuning is pulsed back in reverse steps, where $|\sigma_1 \downarrow_2\uparrow_3\rangle$ returns to $|\sigma_1 S_3\rangle$ in the (102) configuration while $|\sigma_1 \uparrow_2\downarrow_3\rangle$, $|\sigma_1 \uparrow_2\downarrow_3\rangle$, and $|\sigma_1 \downarrow_2\downarrow_3\rangle$ are unloaded to triplet components, $|\sigma_1 T_{23}\rangle$ and $|\sigma_1 T_{\pm 23}\rangle$, staying in (111).

We also carried out the measurement of the electron spin resonance (ESR) signals and the Rabi oscillations to verify the energy spectroscopy and characterize the performance of QD1-3 as spin qubits. The results are shown in Supplementary Fig. 2, which are taken with the pulse sequence in Supplementary Fig. 1c. The spin states are initialized at point I near the (101)–(102) charge transition line in Fig. 2c, where an electron in an excited state can escape to a reservoir and the doubly occupied singlet state $S_3$ is formed. The state $|\sigma_1 S_3\rangle$ is then loaded into $|\sigma_1 \downarrow_2\uparrow_3\rangle$ by the slow adiabatic passage. To drive the electron spin resonance with the micromagnet proximity field[8,9], we applied a microwave burst of duration $t_{\mathrm{burst}} =$

1 μs to the gate electrode shared by the three QDs (horizontal fine gate in Fig. 2b). This leads to the mixing of $|\sigma_1 \downarrow_2 \uparrow_3\rangle$ with $|\sigma_1 \uparrow_2 \uparrow_3\rangle$ ($|\sigma_1 \downarrow_2 \downarrow_3\rangle$) when QD2 (QD3) is in resonance. The DC voltage applied to the shared gate was negatively offset during the microwave burst to prevent leakage of electrons to the reservoirs. Finally, readout is performed at point $M_0$, where the relaxation time of two triplet states, $|T_{+23}\rangle = |\uparrow_2 \uparrow_3\rangle$ and $|T_{-23}\rangle = |\downarrow_2 \downarrow_3\rangle$, is sufficiently long. The two lines are separated by ≈8 mT with the effective g-factor of $|g| = 0.34$, in agreement with $\Delta B_{23}/h = 45$ MHz found in Fig. 4. Similarly, the ESR lines for QD1 and QD2 shown in the upper panel were taken by preparing $|\downarrow_1 \uparrow_2 \sigma_3\rangle$ adiabatically loaded from $|S_1 \sigma_3\rangle$. The Zeeman field difference is found to be ≈18 mT, which is reasonably close to the value of $\Delta B_{12}/h = 45$ MHz used in the main text within the magnitude of the Overhauser field fluctuation. The Rabi oscillations of QD1-3 were taken by repeating the single-shot measurement cycle with $t_{burst}$ increased consecutively from 30 ns to 1.8 μs in 60 steps (Supplementary Fig. 2b). We then calculate the sliding Gaussian average of $P_S$, first against the microwave frequency with the standard deviation of $\sigma_f = 0.2$ MHz, and then against $t_{burst}$ with $\sigma_t = 30$ ns.

**Model of the coherent dynamics**. We describe our spin chain in the Fermi-Hubbard model and the Hamiltonian of the relevant energy levels reads

$$\mathcal{H} = H_t + H_\varepsilon + H_Z$$
$$H_t = \sum_{\sigma_3} (t_{12}|S_1\sigma_3\rangle\langle S_{12}\sigma_3| + \text{h.c.}) + \sum_{\sigma_1} (t_{23}|\sigma_1 S_3\rangle\langle \sigma_1 S_{23}| + \text{h.c.})$$
$$H_\varepsilon = \sum_{\sigma_3} \frac{\varepsilon - \varepsilon_L}{2} |S_1\sigma_3\rangle\langle S_1\sigma_3| + \sum_{\sigma_3} \frac{-\varepsilon + \varepsilon_R}{2} |\sigma_1 S_3\rangle\langle \sigma_1 S_3| \qquad (2)$$
$$H_Z = \sum_i B_i \hat{s}_i^z,$$

where $S_i$ denotes the doubly occupied singlet state in QD$i$ and $\hat{s}_i^z$ is the z component of the spin operator. The local Zeeman energies are given by $B_1 = |g|\mu_B B_{ext} - \Delta B_{12}$, $B_2 = |g|\mu_B B_{ext}$, and $B_3 = |g|\mu_B B_{ext} + \Delta B_{23}$ (note that $B_i$ and $\Delta B_{ij}$ are defined in units of energy) with $g$ the electronic g-factor and $\mu_B$ the Bohr magneton. To explain the experimental results, we find it necessary to take into account the detuning-dependent inter-dot tunnel couplings. Here we adopted phenomenological functions of the forms $t_{12} = t_L \exp\left[-\left(\frac{\varepsilon - \varepsilon_L}{w}\right)^2\right]$ and $t_{23} = t_R \exp\left[-\left(\frac{\varepsilon - \varepsilon_R}{w}\right)^2\right]$ with $w/h = 30.0$ GHz, which are similar to those used in the previous TQD experiments[2,15,34].

The Hamiltonian in Eq. 2 is diagonalized using the program[35] QuTiP. To fit the experimental data in Fig. 4b with calculated energy gaps, we also take into account the $S_1$–$T_{+12}$ and $S_3$–$T_{+23}$ resonance points which appear as faint vertical lines at $\varepsilon_L$–$\varepsilon = B_1 + B_2$ and $\varepsilon$–$\varepsilon_R = B_2 + B_3$ for $t_{evolve} > 100$ ns. These constraints together with the spectral lines in Fig. 4b are sufficient to determine all the unknown parameters in Eq. 2. We also confirmed that the derived energy scale of the detuning axis agrees with the photon-assisted tunneling signal. The calculated eigenenergies for the derived parameters are shown in Fig. 2d and Supplementary Fig. 3.

The coherent spin dynamics of the system is calculated by numerically solving the Lindblad master equation,

$$\dot{\rho}(t) = -\frac{i}{\hbar}[\mathcal{H}(t), \rho(t)]$$
$$+ \sum_n \left[\mathcal{L}_n \rho(t)\mathcal{L}_n^\dagger - \frac{1}{2}\mathcal{L}_n^\dagger \mathcal{L}_n \rho(t) - \frac{1}{2}\rho(t)\mathcal{L}_n^\dagger \mathcal{L}_n\right], \qquad (3)$$

where $\mathcal{L}_n$ are Lindblad operators describing the coupling between the system and the environment. The detuning $\varepsilon$ is varied as a function of time to simulate the approximate pulse waveforms used in the experiment. Among several possible decoherence sources, the spin relaxation is unimportant because the relaxation time is much longer than the submicrosecond timescale of interest. Instead, the fluctuation of the energy levels leads to pure dephasing in the spin dynamics. The energy levels $E_k$ as described by Eq. 2 are susceptible to the charge noise in the electrostatic potentials defining $\varepsilon$, $\varepsilon_{L,R}$, and $t_{L,R}$ as described by Eq. 1. The choice of the ratio $\gamma_{\varepsilon_{L,R}}/\gamma_{t_{L,R}} = 100$ is somewhat arbitrary for the fits in Fig. 5, and it does not affect the conclusion. Here we used $\gamma_{\varepsilon_{L,R}}/\gamma_{t_{L,R}} \gg 1$ for consistency with previous studies[6]. The dephasing-assisted adiabaticity of the spin swap process is clearly visible in the simulation result shown in Supplementary Fig. 4, where the population of $|\downarrow_1 \uparrow_2 \uparrow_3\rangle$ increases monotonically as the dephasing rate increases. We note that the population of $|\uparrow_1 \downarrow_2 \uparrow_3\rangle$ at $\varepsilon = 0$ also increases with the dephasing rate due to the increased adiabaticity across the transition at $\varepsilon = \varepsilon_R$. This suggests that the dephasing noise could also influence the visibility of the standard S–T oscillations observed in double QDs. However, the consequence may be subtler than the TQD case as the effect does not generate peculiar coherent states.

The charge noise in our system most probably originates from the Johnson–Nyquist voltage noise of the room temperature electronics. The noise is fed to gate electrodes through the broadband coaxial lines which attenuate the noise power at 10 GHz only by 12 dB (including the effect of the attenuators and the transmission loss) as measured by the vector network analyzer. The broadband spectrum of this noise contributes the most to the dephasing effect discussed in the main text, while the nuclear Overhauser field is quasi-static, having negligible

impact on the coherent dynamics. The low-frequency part of the voltage noise (below the 10 GHz bandwidth) adds up only to $\delta\varepsilon_{rms} \approx 0.1$ μeV in the root-mean-square fluctuation of the detuning energy, which is well within the noise level commonly observed in similar devices[36]. Although the fluctuations in $\varepsilon$, $\varepsilon_{L,R}$, and $t_{L,R}$ could be correlated with each other due to finite crosstalk, this effect has no significant influence on the analysis.

While the charge noise in our system has no memory effect in the timescale of the spin dynamics (Markovian), the magnetic noise due to the Overhauser field fluctuation is much slower and mainly leads to the ensemble phase averaging effect. The magnetic noise overwhelms the charge noise only when $\Gamma_{kl}$ is reduced for $\varepsilon_L \ll \varepsilon \ll \varepsilon_R$. We take this effect into account separately after the numerical calculation, by making the calculated oscillations decay with a Gaussian envelope of $T_2^* = 60$ ns. Similarly, the slow fluctuations of $\varepsilon$, which could arise from $1/f$-like noise, is taken into account by calculating the convolution with a Gaussian kernel of the standard deviation $\sigma_\varepsilon = 0.9$ GHz.

The simulation result and the experimental data in Fig. 3 are in reasonable agreement for distant entanglement (red line and circles) without assuming an error in the singlet-triplet readout. This reflects the high fidelity (≈90%) of the single-shot readout optimized for the measurement of this regime. On the other hand, we assume a lower readout fidelity of 75% for the triplet outcome in the case of nearest-neighbor entanglement (blue line and circles). The change of this readout fidelity is also found as a change of the visibility near $\varepsilon = 0$ in Fig. 4a. This is probably due to the shift of the measurement point $M_1$ caused by the transient effect of the detuning pulse.

**Data availability**. The data that support the findings of this study are available from the corresponding authors upon reasonable request.

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

## Acknowledgements

We thank S. Miyashita, S. Bartlett, and P. Stano for fruitful discussions. We thank RIKEN CEMS Emergent Matter Science Research Support Team and Microwave Research Group at Caltech for technical assistance. Part of this work was financially supported by the CREST, JST (JPMJCR15N2, JPMJCR1675), the ImPACT Program of Council for Science, Technology, and Innovation (Cabinet Office, Government of Japan), JSPS KAKENHI Grants No. 26220710 and No. 18H01819, and RIKEN Incentive Research Projects. T.O. acknowledges support from JSPS KAKENHI Grants No. 16H00817 and No. 17H05187, PRESTO (JPMJPR16N3), JST, Advanced Technology Institute Research Grant, the Murata Science Foundation Research Grant, Izumi Science and Technology Foundation Research Grant, TEPCO Memorial Foundation Research Grant, The Thermal and Electric Energy Technology Foundation Research Grant, The Telecommunications Advancement Foundation Research Grant, Futaba Electronics Memorial Foundation Research Grant, MST Foundation Research Grant, and Kato Foundation for Promotion of Science Research Grant. A.D.W. and A.L. greatfully acknowledge support from Mercur Pr2013-0001, BMBF Q.Com-H 16KIS0109, TRR160, and DFH/UFA CDFA-05-06. X.H. thanks support by the US ARO. F.N. is partially supported by the MURI Center for Dynamic Magneto-Optics via the AFOSR Award No. FA9550-14-1-0040, the JSPS (KAKENHI), the IMPACT program of JST, CREST Grant No. JPMJCR1676, RIKEN-AIST Challenge Research Fund, JSPS-RFBR Grant No. 17-52-50023, and the Sir John Templeton Foundation.

## Author contributions

T.N. and S.T. planned the project. A.L. and A.D.W. grew the heterostructure and T.N. fabricated the device. T.N. conducted the experiment with the assistance of M.R.D, T.O., J.Y., and A.N.; T.N. analyzed the data and performed numerical calculations with the help of X.H., M.R.D., S.A., J.Y., T.O., and F.N. All authors discussed the results and commented on the manuscript. The project was supervised by S.T.

## Additional information

**Competing interests:** The authors declare no competing interests.

