## [Peer Review File · Nature Communications]

Reviewers' comments:

Reviewer #4 (Remarks to the Author):

In the manuscript, Nakajima et al present quantum oscillation measurements on GaAs gate defined triple quantum dots. Spin qubits are extensively studied subject in GaAs quantum dots and depending on the definition of basis, various types of spin qubits are possible: single spin, singlet-triplet, exchange only, hybrid qubit. etc. Among them the authors' work focus on multiple, single electron spin qubits, that is, spin up-down as qubit basis. Although entanglement between two neighbor spins (singlet – triplet oscillation) was demonstrated in the numerous earlier works, to my knowledge, showing singlet – triplet oscillation between next nearest neighbor spins is the first time, and I acknowledge the novel experiment performed here.

As I became the reviewer of this manuscript in the 2nd round, I would like to give overall evaluation whether the authors properly addressed previous issues of the other referees, as well as my comments to some of them.

1. Minor issues (of all reviewer #1,2,and 3) : All the minor issues such as figure labels, addition of symbol definitions etc. are now correctly defined and explained. I agree with the authors that in the Markovian noise $T2^* = T2$ so that it is appropriate to use $1/T2$ in the Fig. 4b

2. Major points:

A. Entanglement vs. degree of entanglement: I agree with the reviewer #2 that the current manuscript does not measure the degree of entanglement by performing, for example, state tomography on the two distant spins. However, I understand that the authors only claim that the work demonstrates the coherent singlet-triplet oscillations of the distant quantum dot spins. I think this is an issue raised many times in reviewing quantum dot qubit papers. I personally believe that the distant two spins showing coherent singlet – triplet oscillations must be entangled to some extent, and in this respect, the manuscript shows successful transfer of entanglement. On the other hand, if using the term 'entanglement' in the title seems to be unsubstantiated, I can suggest the authors to change the title starting with something like "Coherent singlet triplet oscillations of distant spin qubits ... ". This is merely a suggestion.

B. Adoptability of this technique: The authors argue that, compared to more conventional SWAP operation, the present method can be useful considering timing & decoherence issues related to SWAP operation. I partially agree with the authors on this point. Because I also see that the authors' method must have some restriction on pulse timing (to stay in the adiabatic limit) as well and for this technique to work, leakage can be a major issue, as I already see reduced visibility for distant entanglement data in Fig 3b compared to nearest neighbor. I believe each technique for entanglement shuttling will have its own pros and cons, and I suggest the authors to add a few more sentences in the discussion section discussing using their method in the larger array in the future, especially by comparing with existing method. This will help the readers in the same community.

Overall, the work presents experimentally challenging and novel results on quantum coherence and entanglement performed in the semiconductor quantum dots, and I recommend the manuscript for the publication in Nature Communications. In particular, even though the manuscript does not show measurement of degree of entanglement (nor claim the authors that they did), successful measurement of coherent singlet – triplet oscillations between next nearest neighbor spin is a novel and important result.

Reviewer #5 (Remarks to the Author):

This paper by Takashi Nakajima and colleagues studies a system of three linearly arranged semiconductor quantum dots filled with a total of three electrons in the presence of a magnetic field gradient. An experiment is reported where an entangled singlet state between two single electron spins residing in the two distant quantum dots is prepared. The theoretical model is presented that describes the results and takes into account dephasing noise. It is claimed that entanglement shuttling from an initially localized entangled spin singlet has been achieved, and that this process has been assisted by the dephasing noise.

In contrast to earlier work (Ref. [7]) where two electrons in a triple dot with an empty mediator (center) dot were used, here both the preparation and detection mechanisms are different, as Landau-Zener type avoided crossings with partially adiabatic energy tuning of the quantum dot potentials were used in this work. This method also seems more scalable to larger distances compared to other methods.

This is an impressive novel achievement that may have significant implications for quantum information processing with spin qubits, as well as for other areas of science and technology, as pointed out by the Authors. In this sense I agree with the previous Referees that this work is very important, of broad interest, and of high quality, and that it represents a significant result obtained from a technically challenging experiment. Generally speaking, I find the work suitable for publication in Nature Communications. However, I partially agree with the previous Referees regarding the danger of unsubstantiated claims made in the paper.

Previous Referees have already commented on the solidity of the evidence for the claims made in the paper (already in the title):

- (1) entanglement shuttling from an initially localized entangled spin singlet has been achieved
- (2) this process has been assisted by the dephasing noise

The Authors have replied to both challenges by the Referees. Below I add my own comments.

Regarding (1), I agree with the Authors that there is good evidence for a non-local entangled singlet between dots 1 and 3, provided

by S-T oscillations with a frequency determined by the 1-3 field gradient. This is a promising method for preparation of remote entanglement.

Of course a measurement of entanglement fidelity would be great, but I think it can be accepted to be outside the scope of this work.

Regarding (2), I am not convinced that the dephasing noise increases the entanglement fidelity. This is not so much about "interpretations" of the experiment, but about the meaning of the word "assisted"-- what does dephasing noise really do? Does it directly improve the entanglement, or is the process simply immune against dephasing noise (which would also be a good, albeit different, thing).

I strongly recommend that this issue is clarified before publication.

In addition to this main concern, I have a few minor suggestions and comments which I list below.

(a) The return probability P_S is not explicitly defined in the paper. Since this is a TQD, it might not be

obvious to every reader to which state one is trying to return. I believe it is the (1,0,2) singlet, and

perhaps it is implicitly clear because this is the initial state (hence "return" probability).

Nevertheless,

it would be helpful to say this explicitly when P_S is used for the first time.

(b) Fig, 1a is a bit misleading, because there are three dots in the experiment, not four. I suppose the idea was to show the general principle which should be scalable to more than three dots. But I would at least have expected a "warning" note in the figure caption that says this and declares that in the present experiment there are only three dots.

(c) Related Landau-Zener dynamics have been observed in double quantum dots by the Petta group at Princeton, where multi-slope pulse shapes like those shown in Fig. 1 of the extended data have been used extensively,
J. R. Petta et al., *Science* 327, 669 (2010)
Also the effect of charge noise on such Landau-Zener transitions have been studied in this context,
H. Ribeiro et al. *Phys. Rev. B* 87, 235318 (2013).

(d) typo on line 177: biological (missing 'o')

(e) Ref. [7] has been published as T. A. Baart et al., *Nature Nano.* 12, 26 (2017).

We thank the reviewers for their careful considerations on the manuscript. Below we list our point-by-point responses to all comments from the reviewers. We believe that all the concerns are addressed and our manuscript is now appropriate for publication in *Nature Communications*.

Here is the summary of changes (coloured red in the text):

1. The title is changed.
2. Discussion on the role of the dephasing noise is added in lines 167-172.
3. Discussion on the application to a larger system is added in lines 182-190.
4. Definition of P_S, missing references, and typos are corrected.
5. A data availability statement is added at the end of the Methods section.

Response to Reviewer #4

--

In the manuscript, Nakajima et al present quantum oscillation measurements on GaAs gate defined triple quantum dots. Spin qubits are extensively studied subject in GaAs quantum dots and depending on the definition of basis, various types of spin qubits are possible: single spin, singlet-triplet, exchange only, hybrid qubit. etc. Among them the authors' work focus on multiple, single electron spin qubits, that is, spin up-down as qubit basis. Although entanglement between two neighbor spins (singlet – triplet oscillation) was demonstrated in the numerous earlier works, to my knowledge, showing singlet – triplet oscillation between next nearest neighbor spins is the first time, and I acknowledge the novel experiment performed here.

We thank the reviewer for understanding the novelty of our work.

As I became the reviewer of this manuscript in the 2nd round, I would like to give overall evaluation whether the authors properly addressed previous issues of the other referees, as well as my comments to some of them.

1. *Minor issues (of all reviewer #1,2,and 3) : All the minor issues such as figure labels, addition of symbol definitions etc. are now correctly defined and explained. I agree with the authors that in the Markovian noise $T2^* = T2$ so that it is appropriate to use $1/T2$ in the Fig. 4b*

We appreciate the effort of the reviewer to confirm that all the issues are properly addressed.

2. *Major points:*

A. *Entanglement vs. degree of entanglement: I agree with the reviewer #2 that the current manuscript does not measure the degree of entanglement by performing, for example, state tomography on the two distant spins. However, I understand that the authors only claim that the work demonstrates the coherent singlet-triplet oscillations of the distant quantum dot spins. I think this is an issue raised many times in reviewing quantum dot qubit papers. I personally believe that the distant two spins showing coherent singlet – triplet oscillations must be entangled to some extent, and in this respect, the manuscript shows successful transfer of entanglement. On the other hand, if using the term 'entanglement' in the title seems to be unsubstantiated, I can suggest the authors to change the title starting with something like "Coherent singlet triplet oscillations of distant spin qubits ... ". This is merely a suggestion.*

We thank the reviewer for understanding that the manuscript shows successful transfer of entanglement by observing the coherent singlet-triplet oscillations of the distant quantum dot spins. On the other hand, as the reviewer suggests, some readers might feel that the term 'entanglement' is unsubstantiated. We therefore follow the suggestion of the reviewer and change the title of the manuscript to "*Coherent transfer of electron spin correlations assisted by dephasing noise*".

B. *Adoptability of this technique: The authors argue that, compared to more conventional SWAP operation, the present method can be useful considering timing & decoherence issues related to SWAP*

operation. I partially agree with the authors on this point. Because I also see that the authors' method must have some restriction on pulse timing (to stay in the adiabatic limit) as well and for this technique to work, leakage can be a major issue, as I already see reduced visibility for distant entanglement data in Fig 3b compared to nearest neighbor. I believe each technique for entanglement shuttling will have its own pros and cons, and I suggest the authors to add a few more sentences in the discussion section discussing using their method in the larger array in the future, especially by comparing with existing method. This will help the readers in the same community.

We agree with the reviewer that it is helpful to readers to clarify pros and cons of the present scheme. In short, the use of the dephasing noise makes entanglement shuttling significantly easier to implement, but the lack of the controllability limits the speed and the reliability at present. We have added a discussion on this issue in the revised manuscript (lines 182-190).

Overall, the work presents experimentally challenging and novel results on quantum coherence and entanglement performed in the semiconductor quantum dots, and I recommend the manuscript for the publication in Nature Communications. In particular, even though the manuscript does not show measurement of degree of entanglement (nor claim the authors that they did), successful measurement of coherent singlet – triplet oscillations between next nearest neighbor spin is a novel and important result.

We thank the reviewer very much for finding the importance of our results and recommending the publication in *Nature Communications*. We hope that the remaining issues are also properly addressed in the revised manuscript.

Response to Reviewer #5

--

This paper by Takashi Nakajima and colleagues studies a system of three linearly arranged semiconductor quantum dots filled with a total of three electrons in the presence of a magnetic field gradient. An experiment is reported where an entangled singlet state between two single electron spins residing in the two distant quantum dots is prepared. The theoretical model is presented that describes the results and takes into account dephasing noise. It is claimed that entanglement shuttling from an initially localized entangled spin singlet has been achieved, and that this process has been assisted by the dephasing noise.

In contrast to earlier work (Ref. [7]) where two electrons in a triple dot with an empty mediator (center) dot were used, here both the preparation and detection mechanisms are different, as Landau-Zener type avoided crossings with partially adiabatic energy tuning of the quantum dot potentials were used in this work. This method also seems more scalable to larger distances compared to other methods.

This is an impressive novel achievement that may have significant implications for quantum information processing with spin qubits, as well as for other areas of science and technology, as pointed out by the Authors. In this sense I agree with the previous Referees that this work is very important, of broad interest, and of high quality, and that it represents a significant result obtained from a technically challenging experiment. Generally speaking, I find the work suitable for publication in Nature Communications. However, I partially agree with the previous Referees regarding the danger of unsubstantiated claims made in the paper.

We thank the reviewer for finding the work is very important, of broad interest, and suitable for publication in *Nature Communications* generally speaking.

Previous Referees have already commented on the solidity of the evidence for the claims made in the paper (already in the title):

- (1) entanglement shuttling from an initially localized entangled spin singlet has been achieved*
- (2) this process has been assisted by the dephasing noise*

The Authors have replied to both challenges by the Referees. Below I add my own comments.

Regarding (1), I agree with the Authors that there is good evidence for a non-local entangled singlet between dots 1 and 3, provided by S-T oscillations with a frequency determined by the 1-3 field gradient. This is a promising method for preparation of remote entanglement.

Of course a measurement of entanglement fidelity would be great, but I think it can be accepted to be outside the scope of this work.

We thank the reviewer for recognizing the soundness of our claim.

Regarding (2), I am not convinced that the dephasing noise increases the entanglement fidelity. This is not so much about "interpretations" of the experiment, but about the meaning of the word "assisted"-- what does dephasing noise really do? Does it directly improve the entanglement, or is the process simply immune against dephasing noise (which would also be a good, albeit different, thing).

I strongly recommend that this issue is clarified before publication.

By the word "assisted", we mean that the dephasing noise facilitates the adiabatic spin swap between QD1 and QD2 which is necessary to generate the distant entanglement. The coherence of the generated entangled state is also immune against the noise, but this is a different thing as the reviewer points out. To clarify this issue, we have added a few sentences in the revised manuscript (lines 167-172).

In addition to this main concern, I have a few minor suggestions and comments which I list below.

(a) The return probability P_S is not explicitly defined in the paper. Since this is a TQD, it might not be obvious to every reader to which state one is trying to return. I believe it is the (1,0,2) singlet, and perhaps it is implicitly clear because this is the initial state (hence "return" probability). Nevertheless, it would be helpful to say this explicitly when P_S is used for the first time.

We appreciate the reviewer's helpful suggestion. We have expanded the definition of P_S written in lines 81-83 in the revised manuscript to be more explicit.

(b) Fig. 1a is a bit misleading, because there are three dots in the experiment, not four. I suppose the idea was to show the general principle which should be scalable to more than three dots. But I would at least have expected a "warning" note in the figure caption that says this and declares that in the present experiment there are only three dots.

We agree with the reviewer that a "warning" note is helpful to avoid the confusion. For this purpose, we have added a sentence to the caption of Fig. 1a in the revised manuscript.

(c) Related Landau-Zener dynamics have been observed in double quantum dots by the Petta group at Princeton, where multi-slope pulse shapes like those shown in Fig. 1 of the extended data have been used extensively, J. R. Petta et al., Science 327, 669 (2010)

Also the effect of charge noise on such Landau-Zener transitions have been studied in this context, H. Ribeiro et al. Phys. Rev. B 87, 235318 (2013).

We thank the reviewer for the suggestion. We agree that these works are relevant to ours, especially when discussing the anti-crossing between the singlet and a spin-polarized triplet. Therefore we have added citation to these works at line 116 in the revised manuscript.

(d) typo on line 177: biological (missing 'o')

(e) Ref. [7] has been published as T. A. Baart et al., Nature Nano. 12, 26 (2017).

We thank the reviewer for finding these errors. They have been corrected in the revised manuscript.

REVIEWERS' COMMENTS:

Reviewer #5 (Remarks to the Author):

After having read the Authors' reply to the comments by all reviewers, as well as their changes to the manuscript, I now recommend this work for publication in Nature Communications without any further reservations or comments. I appreciate the Authors' efforts to clarify the role of dephasing, as well as the other improvements and clarifications in their paper.

We thank the reviewer for corroborating all the changes to the manuscript. As there is no remaining issues raised by the reviewers, we believe that our manuscript is now appropriate for publication in *Nature Communications*.

Response to Reviewer #5

--

After having read the Authors' reply to the comments by all reviewers, as well as their changes to the manuscript, I now recommend this work for publication in Nature Communications without any further reservations or comments. I appreciate the Authors' efforts to clarify the role of dephasing, as well as the other improvements and clarifications in their paper.

We thank the reviewer for checking our reply to the comments and all the changes to the manuscript, as well as for helpful suggestions to clarify the manuscript.